# Buffalo Whey-Based Cocoa Beverages with Unconventional Plant-Based Flours: The Effect of Information and Taste on Consumer Perception

Madian Johel Galo Salgado [1,2], Iuri Lima dos Santos Rosario [2], Arlen Carvalho de Oliveira Almeida [2,3], Bruna Samara dos Santos Rekowsky [1,2], Uiara Moreira Paim [1,2], Deborah Murowaniecki Otero [3,4], Maria Eugênia de Oliveira Mamede [3] and Marion Pereira da Costa [1,2,3,*]

[1] Graduate Program in Animal Science in the Tropics (PPGCAT), School of Veterinary Medicine, Federal University of Bahia (UFBA), Ondina, Salvador 40170-110, BA, Brazil; madiangalo16@gmail.com (M.J.G.S.); bruna.rekowsky@hotmail.com (B.S.d.S.R.); uiara.moreira@ufba.br (U.M.P.)

[2] Laboratory of Technology and Inspection of Milk and Derivatives (LaITLácteos), School of Veterinary Medicine, Federal University of Bahia (UFBA), Ondina, Salvador 40170-110, BA, Brazil; iuri_rosario@id.uff.br (I.L.d.S.R.); arlencoa@gmail.com (A.C.d.O.A.)

[3] Graduate Program in Food Science, Faculty of Pharmacy, Federal University of Bahia (UFBA), Ondina, Salvador 40170-115, BA, Brazil; deborah.otero@ufba.br (D.M.O.); mmamede@ufba.br (M.E.d.O.M.)

[4] Graduate Program in Food, Nutrition and Health, Nutrition School, Federal University of Bahia, Campus Canela, Salvador 40110-907, BA, Brazil

[*] Correspondence: marioncosta@ufba.br; Tel.: +55-71-3283-6711

**Abstract:** This study aimed to evaluate the addition of açaí, beetroot, and hibiscus flour on the sensory characteristics of a buffalo whey-based cocoa beverage and, second, to consider if health and sustainability claims could enhance consumer acceptance and purchase intention for the buffalo whey-based cocoa beverage. In this sense, five treatments were elaborated; BCC, the control with a commercial beverage formulation; BCE, the experimental control; BFA, with the addition of açaí flour; BFB, with added beetroot flour; and BFH, with the addition of hibiscus flour. The experiment was divided into two stages: In the first, the beverages were submitted to sensory analyses of acceptance (nine-point hedonic scale), purchase intention and just-about-right (five points), and check-all-that-apply (CATA). In the second stage, the beverages with the highest and lowest acceptance rates were taken, and they were subjected to the effect of sustainability and health information on consumer acceptance, purchase intention, and the CATA test using terms referring to emotions and feelings. The addition of flours decreased the beverage acceptance rate compared to the BCC treatment. The treatments were penalized in aroma and sweet taste. There was no effect on the type of information received by the consumer. Probably, the addition of high cocoa percentages can negatively affect the acceptance of products, as well as the use of flour with bitter flavors, due to the greater acceptance of sweeter products.

**Keywords:** consumer perception; buffalo whey; by-products; chocolate flavor; sustainability; beverage

## 1. Introduction

Whey, a prominent by-product of the dairy industry, holds a significant position in annual dairy production, generating over 60 million liters. In the northeast region of Brazil, the production of coalho cheese stands out, obtained from rennet and matured for 10 days, and has a yield of approximately 1 kilo of coalho cheese per 10 L of milk, which generates approximately 9 L of whey [1]. However, this substantial volume of whey production has been accompanied by growing environmental concern since more than 50% of this valuable resource is traditionally discarded [2]. This wasteful practice has far-reaching environmental implications, ranging from water pollution to greenhouse gas emissions [3]. In light of these environmental challenges, there is a pressing need to reassess the role of

whey in the broader context of sustainability within the dairy industry, which presents an opportunity for sustainable practices in the dairy chain, aligning with concepts such as the circular economy to mitigate environmental pollution [4]. One intriguing avenue that merits exploration is the utilization of buffalo whey, a specific variant with distinct attributes. Buffalo whey, while less commonly discussed in the literature, is an essential by-product of buffalo cheese production, which has a notable presence in certain regions, particularly in parts of Brazil [5]. This unique by-product may offer an alternative perspective on the sustainable use of whey, with potential economic and ecological advantages specific to regions where buffalo farming is prevalent. Therefore, as we consider strategies to mitigate the environmental footprint of the dairy industry, we should also examine how buffalo whey, among other types, can be harnessed efficiently.

The utilization of whey in the development of dairy products, such as beverages, presents a multifaceted advantage encompassing practicality, cost-efficiency, and effectiveness. These whey-based beverages, in addition to their inherent nutritional value, can serve as versatile carriers for an array of valuable bioactive compounds found in their constituent ingredients [6,7]. This approach not only elevates the overall appeal of dairy beverages but also aligns seamlessly with the growing consumer demand for functional and health-enhancing products [8]. For instance, the inclusion of purple plant-based ingredients in whey-based beverages can create dairy products that offer not only unexpected taste but also a valuable dose of antioxidants, potentially contributing to overall well-being.

Cocoa, *Theobroma cacao*, is the most used flavoring in the dairy beverage industry due to its flavor, preferred mainly by children [9]. It can be consumed as white chocolate, milk chocolate, and dark chocolate, containing between 70 and 99% cocoa [10]. It is associated with health benefits such as hypertension, due to its high polyphenol content and antioxidant activity, which has helped to increase the market for chocolates with a higher concentration of cocoa [11]. In addition to cocoa, other plant-based ingredients have been used in the food industry, such as flour.

The açaí agroindustry primarily utilizes the fruit's pulp, which constitutes a more 10% of the fruit, predominantly for the production of frozen and juice pulp. This process generates substantial waste, primarily comprised of seeds and fibers, posing a significant environmental concern. Exploring the biological potential of these açaí agroindustry residues, such as açaí flour, has opened up opportunities for the creation of sustainable biotechnological by-products. Açaí seed flour boasts a distinct chemical composition, leading us to hypothesize that this by-product not only retains the phenolic compounds of açaí but also contains other bioactive compounds. Thus, the addition of açaí flour to dairy beverages could potentially exhibit beneficial effects to consumers [12].

Hibiscus (*Hibiscus sabdariffa* L.) is a wild tropical plant from the Malvaceae family. Due to its flavor, the flowers or calyx of the plant are widely consumed in various forms, including hot or cold juices, jellies, sauces, wine, and spices [13,14]. These flowers are characterized by a high content of bioactive compounds such as polyphenolic acid, flavonoids, and anthocyanins, which exhibit strong antioxidant properties. They are known to stimulate the immune system and increase the production of cytokines, thereby reducing the risk of various cardiovascular diseases, obesity-related conditions, diabetes, and carcinogenic activity in addition to acting as immunomodulators [14–16].

Beetroot (*Beta vulgaris* L.) is renowned for its intense red–violet color, attributed to its high content of betalains, which are secondary metabolic phenolics found in the bulb and stalks of the plant [17–19]. Additionally, betalains exhibit antioxidant and anti-inflammatory properties while inhibiting lipid oxidation [17]. Beetroot is also characterized by its elevated sugar content, including glucose and fructose in smaller quantities, with sucrose as the primary carbohydrate source [18,20]. Moreover, it is rich in vitamins A, B1, B2, B5, and C, as well as minerals like potassium, sodium, phosphorus, calcium, zinc, iron, and manganese. It has found application as a coloring agent in yogurt production [18], a sweetener [17], an ingredient in milk ice cream [19], and a color stabilizer in fluid milk [21].

First, we have hibiscus flour, widely consumed in various forms such as tea, juice, and supplements. This natural ingredient is notably rich in anthocyanins, a class of flavonoids renowned for their antioxidant properties [22]. Furthermore, the açaí berry, a popular fruit known for its impressive array of health-promoting components, includes minerals, dietary fibers, phenolic compounds, flavonoids, tocopherols, and essential fatty acids such as linoleic and linolenic acid [23]. Furthermore, beetroot and its byproducts have emerged as promising options for individuals seeking to manage various metabolic disorders like hypertension, diabetes, insulin resistance, and kidney dysfunction [8]. Beetroot's therapeutic potential is attributed to its bioactive compounds, which have been studied for their positive effects on health. Therefore, incorporating hibiscus flour, açaí flour, and beetroot flour into whey-based beverages provides consumers with an accessible and enjoyable means of including these potentially beneficial compounds in their diets. This synergy between whey and various health-promoting ingredients underscores the versatility and potential of dairy beverages in promoting overall well-being.

The terms "sustainability" and "health" are concepts that are gradually becoming clearer to consumers. Consumers often associate sustainability with factors like reducing carbon footprints, avoiding preservatives, treating animals ethically, and using more natural ingredients. From their perspective, sustainability reflects a commitment to environmental and ethical considerations. Conversely, the industry tends to approach sustainability more from an agricultural perspective, focusing on sustainable farming practices and resource management [24]. This difference in interpretation highlights the need for clearer communication between consumers and the food industry regarding the concept of sustainability. Additionally, industries have introduced labeling systems aiming to simplify nutrition and health claims, recognizing that nutritional characteristics can be challenging for the average consumer to understand [25,26]. These labels offer a convenient way for consumers to make informed food choices, providing straightforward guidance on the nutritional content and potential health benefits of products. Through demystifying complex nutritional information, these labels empower consumers to make healthier food choices, promoting overall well-being.

However, it is crucial to strike a balance between sustainability, health, and consumer preferences when developing food products. Neglecting the sensory expectations and preferences of consumers can result in the introduction of products with limited market acceptance [27]. Therefore, during the product development phase, it is imperative to conduct sensory evaluations to gauge how consumers perceive the sensory characteristics of the products. Additionally, consumer acceptance is influenced not only by sensory attributes but also by non-sensory factors such as the brand's reputation, pricing, nutritional composition, and perceived health benefits [28–31].

In this context, the primary objective of this study was to assess the impact of incorporating açaí, beetroot, and hibiscus flour into a 100% cocoa buffalo whey-based beverage on its sensory characteristics. Understanding how these additions affect the product's sensory profile is crucial for ensuring that the beverage aligns with consumer expectations and preferences. Additionally, we aimed to investigate whether emphasizing health and sustainability claims could enhance consumer acceptance, purchase intention, and emotional responses to the buffalo whey-based cocoa beverage. In an era where consumers are increasingly conscious of their health and environmental impact, it is vital to explore how these factors influence their choices. Through highlighting the potential health benefits and sustainable aspects of the beverages, we aimed to determine whether these attributes could positively impact consumers' willingness to accept the product and their overall satisfaction.

## 2. Materials and Methods

### 2.1. Materials

The cow and buffalo milk whey was obtained using the coalho cheese production methodology [1]. The proximal composition of cow and buffalo whey is described in

Table S1. Following extraction, the liquid whey underwent a filtration process, using fabric with micropores that prevented the passage of large curd particles, and was subsequently pasteurized at 65 °C for 30 min. It was then carefully refrigerated at 4 ± 2 °C until the beverage production phase. Açaí flour (Viva Nutureza Ind. de Produtos Naturais Ltda, Viva Nutureza®, Bahia, Brazil), xanthan gum (Sabor Leve, Leve Croc®, Parana, Brazil), xylitol (Natural Vitta Comercio De Variedades Ltda, Natural Vitta®, Bahia, Brazil), and cocoa powder (50% and 100%) (Nestlé Nordeste Alimentos e Bebidas Ltda, Bahia, Brazil) were purchased from a local natural product store in Salvador, Bahia, Brazil. Additionally, inulin (Neovita Foods Ltda, São Paulo, Brazil), beetroot powder (Natural Vitta Comercio De Variedades Ltda, Natural Vitta®, Bahia, Brazil), and hibiscus flour (Della Terra Comercio E Distribuidora De Produtos Naturais E Suplementos Ltda, Della Terra®, São Paulo, Brazil) were acquired from trusted online stores.

### 2.2. Cow and Buffalo Whey-Based Cocoa Beverage Preparation Processing

The cow and buffalo whey-based cocoa beverage was processed at the Laboratory of Inspection and Technology of Milk and Derivatives at the Federal University of Bahia (LaITLácteos/UFBA). Five distinct variations of the cow and buffalo whey-based cocoa beverage were developed, including two types of control: a control variant reflecting the typical characteristics of chocolate beverages found in the local market (BCC) and a control variant with a new formulation (BCE). The remaining three variants consisted of a beverage with açaí flour (BFA), a beverage with beetroot flour (BFB), and a beverage with hibiscus flour (BFH). In the BCE, BFA, BFB, and BFH treatments, xylitol (3.5% *w/v*), inulin (2.5% *w/v*), 100% cocoa powder (7.5% *w/v*), and the respective flour (2.5% *w/v*) were incorporated and mixed with buffalo whey, 86.5% for BCE and 84.0% for BFA, BFB, and BFH, respectively. For the production of the commercial control (BCC), crystal sugar (3.5% *w/v*), xanthan gum (0.25% *w/v*), and 50% cocoa powder (7.5% *w/v*) were added and mixed with 88.75% cow whey. The resulting blends were formulated following good manufacturing practices and homogenized using a food processor (Philips Walita® RI7630, 600 W, Philips Do Brasil Ltda, São Paulo, Brazil). Subsequently, the buffalo whey-based beverages were packaged in sterile glass bottles (2000 mL) and refrigerated at 4 ± 2 °C until the next day to carry out sensory analysis. The beverages were subjected to microbiological analysis according to Brasil (2005) [32] to ensure the hygiene of the final product.

### 2.3. Sensory Acception and Characterization

All sensory evaluations underwent review and approval by the Ethics and Research Committee of UFBA under protocol CAAE 60414022.7.0000.5531. Written informed consent was obtained from all participants during the food-tasting sessions. Individuals with lactose intolerance or allergies to milk and its derivatives were excluded from participating in this research. This study was divided into two sections of sensory research (Figure S1). In the first section, the acceptability and characterization of the cow and buffalo whey-based cocoa beverage were evaluated through overall acceptability, purchase intention, and JAR (just-about-right) scaling. This was conducted at the School of Veterinary Medicine of the Federal University of Bahia, Brazil, and involved a sample size of 120 untrained participants (84 women and 36 men, respectively), with ages ranging from 18 to 66 years old. In the second study, an assessment of the effect of information was conducted. The analyses were conducted at the Faculty of Pharmacy, Nutrition School, and School of Veterinary Medicine of the Federal University of Bahia, Brazil. The study involved a sample size of 164 untrained participants (114 women and 50 men, respectively), with ages ranging from 18 to 66 years old.

Consumers were randomly recruited based on their interest and willingness to participate in the study. The sensory tests were conducted in individual booths with white lighting, set up in a controlled environment maintained at a temperature between 22 and 24 °C with proper air circulation. The samples, served in 50 mL plastic cups, were chilled to 4 ± 2 °C, and 20 mL of each sample was provided to the assessors. The samples were coded

with 3-digit random codes in sequential monadic order following a balanced complete block design. To cleanse the palate between samples, assessors were provided with salt biscuits and filtered water at room temperature (25 °C).

### 2.3.1. Acceptability and Purchase Intention

In the acceptability test, the assessors were asked to evaluate each sample's appearance, color, odor, flavor, consistency, mouthfeel, and overall impression using a 9-point hedonic scale (from 1 = extremely dislike to 9 = extremely like), using the Table S3 to describe each attribute when needed. Purchase intention was evaluated using a 5-point category scale (1 = certainly would not buy, 5 = certainly would buy) following the methodology of Meilgaard [33].

### 2.3.2. Just-About-Right (JAR)

The JAR test was conducted to determine the optimal intensity of sensory attributes in the cow and buffalo whey-based cocoa beverage. Consumers evaluated attributes such as aroma (sweet, cocoa, bitter, milk), taste (sweet, bitter, sour), flavor (cocoa, whey, caramel), color (amber, caramel, red), and texture (sandiness, consistency, viscosity, mouthfeel) using a just-about-right scale with five points (1 = not enough, 3 = ideal, 5 = too much), as described by Meilgaard [33].

### 2.3.3. Check-All-That-Apply (CATA)

Participants received a list of 24 terms in the check-all-that-apply (CATA) format. Each participant was asked to select all the terms that described the sensory characteristics of each treatment. The terms included attributes related to aroma (fruity aroma, cocoa aroma, whey aroma, sweet aroma, milk aroma, bitter aroma), taste and flavor (bitter taste, sour taste, sweet taste, milk flavor, whey flavor, chocolate flavor, fruity flavor), appearance (dark brown color, light brown color, red color, homogeneous appearance, shiny), and texture (sandiness, fat sensation, foam, consistent, viscous, fluid). Ref. [34] also balanced the presentation order of descriptive terms.

### *2.4. Information Effect on Consumer Perceptions*

The consumer perception test included the participation of 164 untrained assessors. The environmental conditions and sample delivery followed the same procedures as in the previous study. Three samples were used: BFA and BFH, selected based on their acceptance in the previous study, and a dummy sample containing a mixture of both, according to [35]. To mitigate the first-sample effect, the dummy sample was served first, while the other samples were served in a balanced complete block design and sequential monadic order. Only the results of BFA and BFH were considered for the study.

The consumers were randomly divided into three groups: The blind group (62 participants) received no information. The second group (62 participants) received the following information: "You will taste a new cocoa beverage with the addition of purple flours with a high content of antioxidants. The purple flours used in this study contain significant levels of phenolic acids, flavonoids, and anthocyanins, exhibiting antihypertensive, hypoglycemic, hypolipidemic, antianemia, anti-inflammatory, and antioxidant effects, in addition to potential effects on body weight control and loss." The third group (62 participants) received the information: "You will taste a new cocoa beverage made with whey, which is the largest by-product of the dairy industry. Whey is a by-product obtained during cheese production (an average of 9 L of whey is produced for each kilogram of cheese). However, in Brazil, only 60% of this whey is reused. A significant amount is still discarded in nature, posing a high polluting potential, which can harm aquatic life and ecosystems".

Acceptance evaluation included overall taste, bitter taste, mouthfeel, and overall impression as attributes, rated on a 9-point hedonic scale (1 = extremely dislike, 9 = extremely like). Purchase intention was evaluated using a 5-point category scale (1 = certainly would not buy, 5 = certainly would buy). Additionally, a CATA questionnaire consisting

of 18 terms was used to capture responses related to emotions and feelings. The terms included descriptors such as good for health, tasteless, sad, pacific, environmentally friendly, happy, nutritious drink, animal welfare, satisfaction, sustainability, animal exploitation, natural drink, ecological, disappointment, optimism, rejection, energetic, and emotionless.

*2.5. Statistical Analysis*

The acceptance, purchase intention, and consumer perception results were analyzed via one-way ANOVA and subjected to Tukey's test at $p < 0.05$. Penalty analysis was carried out on JAR data to relate the adequacy of the attributes and overall liking where consumers rated the attributes at "not enough" or "too much". In both CATA analyses, the frequency of each descriptor was calculated through counting the number of consumers who used it to describe each sample and using Cochran's Q test ($p < 0.05$) to identify differences between the sample for each descriptor. Principal component analysis (PCA) was performed to obtain a bidimensional representation of the relationship between samples and descriptors. All analyses were performed using the statistical program XLSTAT version 2013.2.03 (Addinsoft, Paris, France).

## 3. Results

*3.1. Acceptance Test and Purchase Intention*

The results of sensory attribute acceptance for the buffalo whey-based cocoa beverage are illustrated in Figure 1. In terms of acceptance, for a product to be considered accepted, it must have an acceptance greater than or equal to seven; using this logic, only the BCC could be considered as accepted [36]. The BCC treatment, formulated to resemble a commercial product, obtained higher acceptance scores in all attributes due to its familiar characteristics for consumers [37], being rated between "slightly like" and "moderately like" (6–7). On the other hand, when comparing treatments with the same formulation (BCE, BFA, BFB, and BFH), significant differences between samples ($p \leq 0.05$) were detected for all attributes, indicating that the addition of açaí, beetroot, and hibiscus flour reduced the acceptance of cocoa buffalo whey milk-based beverages. However, studies [38] have shown that the use of dark chocolate with high concentrations in the production of beverages reduces their acceptance, mainly due to its bitter taste and astringency on the palate, due its high content of alkaloids, amino acids, peptides, pyrazines, and polyphenols, respectively.

However, in terms of appearance and color attributes, which are assessed via sight, there was no significant difference ($p \leq 0.05$) between the BCC, BCE, and BFH treatments. This may be associated with the higher acceptance of foods with more intense color [39]. For instance, adding açaí and beetroot flour resulted in lower ($p < 0.05$) scores for appearance and color; this may have influenced the color of the drinks. Beetroot has a high content of betalains, water-soluble compounds, which could have given the drinks a more purple color [40]; in turn, açaí contains high content of anthocyanins [41]. However, when comparing BFA with BCE, the control with the same formulation, there were no significant ($p > 0.05$) differences in the scores for flavor, consistency, mouthfeel, and overall liking, which were rated between "dislike slightly" and "like slightly" (4–6). In contrast, adding beetroot and hibiscus flour had a pronounced ($p < 0.05$) impact on these attributes.

Here, the influence of the flavor attribute on the acceptance of subsequent attributes or even and the acceptances of the product was observed. Overall, the assessors evaluated the BCC treatment more positively. In contrast, the BFH treatment scored between "dislike very much" and "dislike slightly" (2–4) for the attributes of flavor, consistency, mouthfeel, and overall liking. Notably, the positive evaluation of the BCC treatment contrasted with the less favorable perception of the BFH treatment, emphasizing the intricate relationship between flavor and the overall liking of the beverage [39].

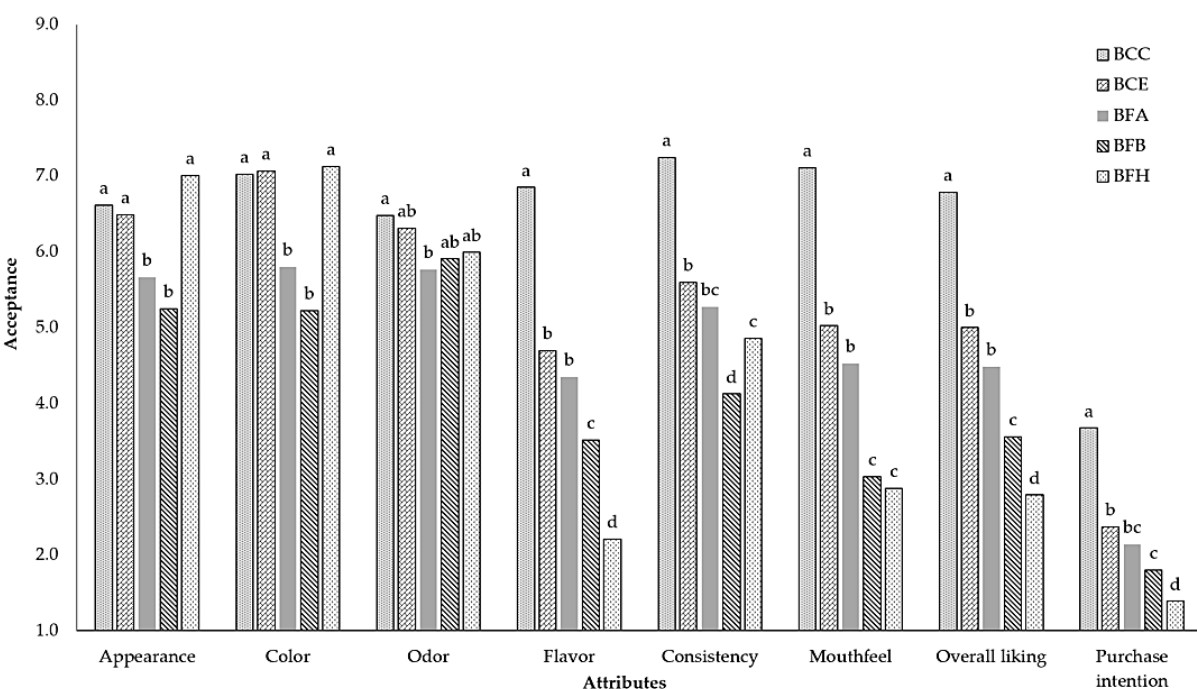

**Figure 1.** The average of the hedonic values obtained from the acceptance test for buffalo whey-based cocoa beverages. [a,b,c,d] Means in the bar followed by different lowercase letters significantly differ from the treatments (Tukey test; $p < 0.05$). BCC = control commercial formulation; BCE = control experimental formulation; BFA = beverage with açaí flour; BFB = beverage with beetroot flour; BFH = beverage with hibiscus flour. Purchase intention analysis was evaluated on a 5-point category scale.

Regarding purchase intention, three groups were observed. The BCC treatment exhibited the highest potential for consumer purchase, with evaluations indicating a likelihood of "might or might not buy" (scores of around three). Conversely, the second group included the BCE and BFA treatments, which garnered responses indicating lower likelihoods, leaning towards "probably would not buy" (scores around two). Notably, the BFH treatment received the least favorable evaluation, with assessors expressing a clear inclination of "definitely would not buy" (scores around one).

### 3.2. Just-About-Right (JAR) and Penalty Analysis

Information for the JAR test and penalty are described in Tables 1 and 2. For the JAR test, a scale of five was used, with three being the ideal point. All treatments are ideal in amber, brown, and red colors (Table 1). However, the BFB treatment was above ideal in red color, due to its betalain content [41]. The BFA, BFB, and BFH treatments had lower scores for amber, brown, and red colors. In the sweet aroma, all treatments were below ideal, with BFH having the greatest effect, which is also reflected in the result of the acid aroma; BFH had a higher score compared to the other treatments. In a similar study, Iwalokun and Shittu [42] observed reduced acceptance of treatments containing hibiscus extract, which is known for its high content of anthocyanins, ascorbic acid, and antioxidants, contributing to an intense bitter taste [43]. However, as expected, the treatments with 100% cocoa had a higher score in the cocoa aroma and flavor attributes; however, despite having the same formulation, the hibiscus treatment was classified as having a lower score in these attributes. In other ways, the aroma of milk was classified as ideal, due to consumers relating the dairy drink directly to milk; however, as can be seen in Table S2, the treatments do not contain milk, demonstrating that replacing milk with whey can be a reliable alternative.

**Table 1.** Just-about-right (JAR) profile scores for the different formulations evaluated.

| Attributes | BCC [1] | BCE | BFA | BFB | BFH |
|---|---|---|---|---|---|
| Amber color | 3.02 ± 0.72 ᵃ | 2.91 ± 0.69 ᵃ | 3.04 ± 0.80 ᵃ | 2.85 ± 0.96 ᵃ | 2.89 ± 0.72 ᵃ |
| Brown color | 2.97 ± 0.70 ᵇᶜ | 3.11 ± 0.71 ᵃᵇ | 3.39 ± 0.91 ᵃᵇ | 2.73 ± 0.94 ᶜ | 3.26 ± 0.74 ᵃ |
| Red color | 2.78 ± 0.72 ᵇ | 2.79 ± 0.74 ᵇ | 3.00 ± 0.94 ᵃᵇ | 3.93 ± 1.06 ᵃ | 2.96 ± 0.78 ᵃᵇ |
| Sweet aroma | 2.89 ± 0.83 ᵃ | 2.44 ± 0.91 ᵇ | 2.29 ± 0.88 ᵇᶜ | 2.27 ± 0.92 ᵇᶜ | 1.98 ± 0.89 ᶜ |
| Cocoa aroma | 2.85 ± 0.78 ᶜ | 3.30 ± 0.89 ᵃ | 3.33 ± 0.91 ᵃ | 3.23 ± 1.05 ᵃᵇ | 2.89 ± 1.21 ᵇᶜ |
| Bitter aroma | 2.93 ± 0.55 ᶜ | 3.17 ± 0.74 ᵇᶜ | 3.29 ± 0.84 ᵇ | 3.32 ± 0.97 ᵇ | 3.80 ± 1.07 ᵃ |
| Milk aroma | 3.05 ± 0.77 ᵃ | 2.59 ± 0.78 ᵇ | 2.45 ± 0.90 ᵇ | 2.44 ± 0.88 ᵇ | 2.37 ± 1.01 ᵇ |
| Sweet taste | 2.90 ± 0.79 ᵃ | 1.96 ± 0.83 ᵇ | 2.10 ± 0.91 ᵇ | 1.93 ± 0.85 ᵇ | 1.45 ± 0.70 ᶜ |
| Bitter taste | 3.00 ± 0.50 ᵇ | 3.24 ± 0.80 ᵇ | 3.33 ± 0.89 ᵇ | 3.25 ± 1.08 ᵇ | 4.25 ± 1.20 ᵃ |
| Sour taste | 3.20 ± 0.69 ᵇ | 3.63 ± 1.08 ᵃ | 3.56 ± 1.05 ᵃᵇ | 3.63 ± 1.14 ᵃ | 3.44 ± 1.40 ᵃᵇ |
| Cocoa flavor | 3.09 ± 0.70 ᵇ | 3.43 ± 0.91 ᵃ | 3.31 ± 0.92 ᵃᵇ | 3.28 ± 1.04 ᵃᵇ | 2.79 ± 1.26 ᶜ |
| Whey flavor | 3.20 ± 0.66 ᵃ | 2.95 ± 0.81 ᵇ | 2.98 ± 0.97 ᵃᵇ | 2.86 ± 1.02 ᵇ | 2.78 ± 1.19 ᵇ |
| Caramel flavor | 2.83 ± 0.71 ᵃ | 2.23 ± 0.81 ᵇ | 2.22 ± 0.89 ᵇ | 2.08 ± 0.93 ᵇ | 1.84 ± 0.95 ᶜ |
| Sandiness | 3.21 ± 0.62 ᶜ | 3.30 ± 0.77 ᵇᶜ | 3.60 ± 1.06 ᵇ | 4.15 ± 1.32 ᵃ | 3.23 ± 0.93 ᶜ |
| Consistency | 3.23 ± 0.73 ᵃ | 2.40 ± 0.78 ᵇ | 2.52 ± 0.82 ᵇ | 2.53 ± 0.99 ᵇ | 2.38 ± 0.80 ᵇ |
| Viscosity | 3.38 ± 0.77 ᵃ | 2.51 ± 0.81 ᵇ | 2.61 ± 0.75 ᵇ | 2.59 ± 0.99 ᵇ | 2.48 ± 0.82 ᵇ |

ᵃ,ᵇ,ᶜ Means in the column followed by different lowercase letters differ from the treatments (Tukey test; $p < 0.05$).
[1] BCC = control commercial formulation; BCE = control experimental formulation; BFA = beverage with açaí flour; BFB = beverage with beetroot flour; BFH = beverage with hibiscus flour.

Treatments with 100% cocoa were classified as non-ideal in terms of sweet flavor; this outcome can be attributed to the high concentration of cocoa in the formulations, which typically results in a more intense chocolate flavor with pronounced acidity and bitterness, as noted by de Jesus Silva et al. [44]. Again, the BFH treatment had a greater effect on the sweet flavor and sour flavor attributes, being suboptimal and above ideal, respectively. The treatment with the addition of beetroot, BFB, was classified above the right level in terms of sandiness, which may be related to the low solubility of beetroot, leaving the sensation of larger particles. Treatments with added inulin were considered lower than the right level compared to treatments with added xanthan gum. Studies have demonstrated the great capacity of xanthan gum to improve viscosity, gel formation, and beverage consistency [45].

In this sense, as expected, regarding sweet aroma and sweet taste, all treatments received penalties for being perceived as insufficient (Table 2); however, treatments containing 100% cocoa were heavily penalized, which demonstrates the low acceptance of dark chocolate [38]. However, the absence of a sweet flavor had a more pronounced impact on the acceptance of the BFH treatment, as it was penalized by a higher percentage of consumers (91.67%) for lacking sufficient sweetness. Similarly, 80.67% of consumers penalized the same treatment for having an excessively bitter taste. However, the effect was also observed in treatments BCE, BFA, and BFB, which were penalized for having an excessive bitter aroma and bitter taste. The BFA, BFB, and BFH treatments received penalties from over 20% of consumers for the whey flavor attribute, indicating individual taste variability [46]. Similar patterns were observed for the attributes of sour taste and cocoa flavor in the BFH treatment, bitter taste in the BFB treatment, and amber color in the BFA treatment. Additionally, consumers noted excessive sandiness in the BCE, BFA, and BFB treatments. In contrast, treatment BCE was considered to have the ideal consistency (Table 2), making it the only one without penalties for this attribute. Only the BCE and BFA treatments were penalized for not having enough viscosity.

**Table 2.** Consumer penalty analysis of the just-about-right (JAR) diagnostic attributes, percentage of consumers (%), and mean drop.

| Attributes | BCC [1] | | BCE | | BFA | | BFB | | BFH | |
|---|---|---|---|---|---|---|---|---|---|---|
| | NE [2] | TM [3] | NE | TM | NE | TM | NE | TM | NE | TM |
| Amber color | ——[5] | —— | —— | —— | 21.67 (0.91)[4] | 22.69 (0.69) | 31.67 (0.50) | —— | —— | —— |
| Brown color | 20.83 (0.56) | —— | —— | —— | —— | 42.86 (0.92) | 39.17 (1.22) | —— | —— | 29.41 (0.71) |
| Red color | —— | —— | —— | —— | 21,01 (1.05) | —— | —— | 70,59 (1.81) | —— | —— |
| Sweet aroma | 26.67 (1.00) | —— | 51.67 (1.41) | —— | 56.67 (2.29) | —— | 52.50 (1.60) | —— | 68.33 (1.33) | —— |
| Cocoa aroma | 26.67 (1.28) | —— | —— | 35.29 (1.74) | —— | 35.29 (1.36) | —— | 32.77 (1.22) | 33.33 (1.53) | —— |
| Bitter aroma | —— | —— | —— | 22.69 (1.64) | —— | 29.41 (1.35) | —— | 36.13 (0.82) | —— | 61.34 (1.70) |
| Milk aroma | —— | —— | 38.33 (1.23) | —— | 45.83 (1.84) | —— | 45.83 (1.75) | —— | —— | —— |
| Sweet taste | 25.00 (1.02) | —— | 74.17 (1.60) | —— | 67.50 (2.20) | —— | 72.50 (1.98) | —— | 91.67 (3.31) | —— |
| Bitter taste | —— | —— | —— | 32.77 (1.28) | —— | 36.13 (1.50) | 20.00 (0.67) | 39.50 (0.68) | —— | 80.67 (2.61) |
| Sour taste | —— | 22.69 (0.81) | —— | 55.46 (2.06) | —— | 51.26 (1.68) | —— | 57.14 (1.70) | 24.17 (2.39) | 53. 78 (1.53) |
| Cocoa flavor | —— | —— | —— | 41.18 (2.47) | —— | 38.66 (1.55) | —— | 36.13 (1.39) | 42.50 (1.32) | 28.57 (0.78) |
| Whey flavor | —— | 20.17 (0.89) | 25.00 (1.87) | —— | 23.33 (2.39) | 25.21 (1.25) | 32.50 (1.63) | 23.53 (1.32) | 42.50 (1.49) | 27.73 (1.38) |
| Caramel flavor | 25.83 (1.06) | —— | 56.67 (1.22) | —— | 57.50 (1.88) | —— | 64.17 (1.09) | —— | 76.67 (0.74) | —— |
| Sandiness | —— | —— | —— | 31.93 (0.56) | —— | 60.50 (1.02) | —— | 79.83 (1.33) | —— | —— |
| Consistency | —— | —— | 48.33 (1.05) | —— | 40.33 (1.27) | —— | 45.83 (0.62) | —— | 50.00 (0.68) | —— |
| Viscosity | —— | —— | 46.67 (1.03) | —— | 38.33 (1.22) | —— | —— | —— | —— | —— |

[1] BCC = control commercial formulation; BCE = control experiment formulation; BFA = beverage with açaí flour; BFB = beverage with beetroot flour; BFH = beverage with hibiscus flour. [2] NE = not enough. [3] TM = too much. [4] The number between the parentheses indicate the mean drop, calculated via subtracting the mean overall JAR acceptance of the group from the mean global acceptance of the group. [5]—indicates that less than 20% of consumers found the sensory attribute from the JAR score to be too little or too much.

### 3.3. Check-All-That-Apply (CATA)

According to the results of the Q Cochran analysis, there were no significant differences ($p < 0.05$) in 5 terms (fruity aroma and flavor, whey aroma and flavor, and milk aroma) out of the 23 terms used in the CATA question to describe the samples (Table 3). These findings, in line with those of Vidal et al. [47], indicate that consumers could perceive differences in the sensory characteristics of the beverages. The CATA method also helps identify the characteristics that impact overall liking [48].

**Table 3.** Frequency (%) of each term of the CATA was indicated by consumers for different formulations of buffalo whey-based cocoa beverage.

| Attribute | BCC [1] | BCE | BFA | BFB | BFH | *p*-Value |
|---|---|---|---|---|---|---|
| Fruity aroma | 6.7 [a] | 9.2 [a] | 5.8 [a] | 13.3 [a] | 12.5 [a] | 0.116 |
| Cocoa aroma | 44.2 [c] | 69.2 [a] | 65.0 [ab] | 52.5 [bc] | 44.2 [c] | <0.0001 |
| Whey aroma | 15.8 [a] | 6.7 [a] | 15.0 [a] | 14.2 [a] | 10.0 [a] | 0.053 |
| Sweet aroma | 32.5 [a] | 11.7 [b] | 10.8 [b] | 10.8 [b] | 5.0 [b] | <0.0001 |
| Milk aroma | 11.7 [a] | 7.5 [ab] | 5.0 [ab] | 1.7 [b] | 4.2 [ab] | 0.012 |
| Bitter aroma | 0.8 [c] | 12.5 [bc] | 20.8 [b] | 10.0 [bc] | 40.0 [a] | <0.0001 |
| Bitter taste | 1.7 [c] | 29.2 [b] | 35.0 [b] | 35.0 [b] | 81.7 [a] | <0.0001 |
| Sour taste | 19.2 [c] | 75.8 [a] | 69.2 [ab] | 66.7 [ab] | 57.5 [b] | <0.0001 |
| Sweet taste | 49.2 [a] | 5.8 [b] | 8.3 [b] | 5.8 [b] | 0.8 [b] | <0.0001 |
| Milk flavor | 40.8 [a] | 8.3 [b] | 11.7 [b] | 5.8 [b] | 3.3 [b] | <0.0001 |
| Whey flavor | 11.7 [a] | 14.2 [a] | 19.2 [a] | 10.8 [a] | 14.2 [a] | 0.250 |
| Chocolate flavor | 66.7 [a] | 40.0 [b] | 29.2 [b] | 25.8 [bc] | 10.8 [c] | <0.0001 |
| Fruity flavor | 3.3 [a] | 5.8 [a] | 7.5 [a] | 10.8 [a] | 11.7 [a] | 0.045 |
| Dark brown color | 23.3 [d] | 54.2 [bc] | 80.8 [a] | 39.2 [cd] | 70.0 [ab] | <0.0001 |
| Light brown color | 50.8 [a] | 28.3 [b] | 5.0 [c] | 6.7 [c] | 2.5 [c] | <0.0001 |
| Red color | 5.0 [b] | 6.7 [b] | 15.8 [b] | 68.3 [a] | 17.5 [b] | <0.0001 |
| Sandiness | 16.7 [c] | 30.0 [c] | 60.0 [b] | 79.2 [a] | 26.7 [c] | <0.0001 |
| Fat sensation | 17.5 [a] | 5.8 [c] | 13.3 [bc] | 10.0 [bc] | 5.8 [c] | 0.007 |
| Foam | 32.5 [a] | 9.2 [bc] | 17.5 [b] | 4.2 [c] | 3.3 [c] | <0.0001 |
| Homogeneous appearance | 46.7 [ab] | 45.8 [ab] | 30.0 [b] | 30.8 [b] | 49.2 [a] | 0.001 |
| Shiny | 63.3 [a] | 61.7 [a] | 48.3 [ab] | 43.3 [b] | 57.5 [ab] | 0.001 |
| Consistent | 49.2 [a] | 2.5 [b] | 5.0 [b] | 5.8 [b] | 3.3 [b] | <0.0001 |
| Viscosity | 44.2 [a] | 3.3 [b] | 5.0 [b] | 11.7 [b] | 2.5 [b] | <0.0001 |
| Fluid | 19.2 [d] | 51.7 [a] | 38.3 [ab] | 20.8 [cd] | 35.8 [bc] | <0.0001 |

[a,b,c,d] Means in the same line followed by different lowercase letters differ from the treatments (Tukey test; $p < 0.05$). [1] BCC = control commercial formulation; BCE = control experiment formulation; BFA = beverage with açaí flour; BFB = beverage with beetroot flour; BFH = beverage with hibiscus flour.

Consequently, terms such as "sour taste" and "dark brown color" were frequently used to describe the treatments containing 100% cocoa, which confirms the results obtained from the penalty analysis (Table 2). In addition, the BCC treatment was predominantly described using terms such as "sweet taste", "milk flavor", "chocolate flavor", "light brown color", "consistency", and "viscosity" when compared to the other treatments. These terms can be considered the main characteristics contributing to increased product acceptance. Furthermore, the BFH treatment was mostly associated with the terms "sour aroma" (40.0%) and "sour taste" (81.7%).

Figure 2 depicts a two-dimensional representation of the principal component analysis (PCA) conducted on the CATA data of buffalo whey-based cocoa beverages. The PCA analysis accounted for 84.47% of the variation in the data, with the first and second dimensions explaining 68.24% and 19.23% of the variance, respectively. Among the various treatments, the BCC treatments showed the highest number of associated terms, including "milk aroma", "foam", "light brown color", "milk flavor", "chocolate taste", "sweet aroma", "viscosity", "consistency", and "sweet taste." In contrast, the BCE, BFA, and BFH treatments were primarily linked to terms like "sour taste", "sour aroma", "fluidity", "dark brown color", "cocoa aroma", "bitter taste", "fruity flavor", "homogeneous appearance", and "shininess". Additionally, the BFB treatment was predominantly associated with "sandiness" and "red color".

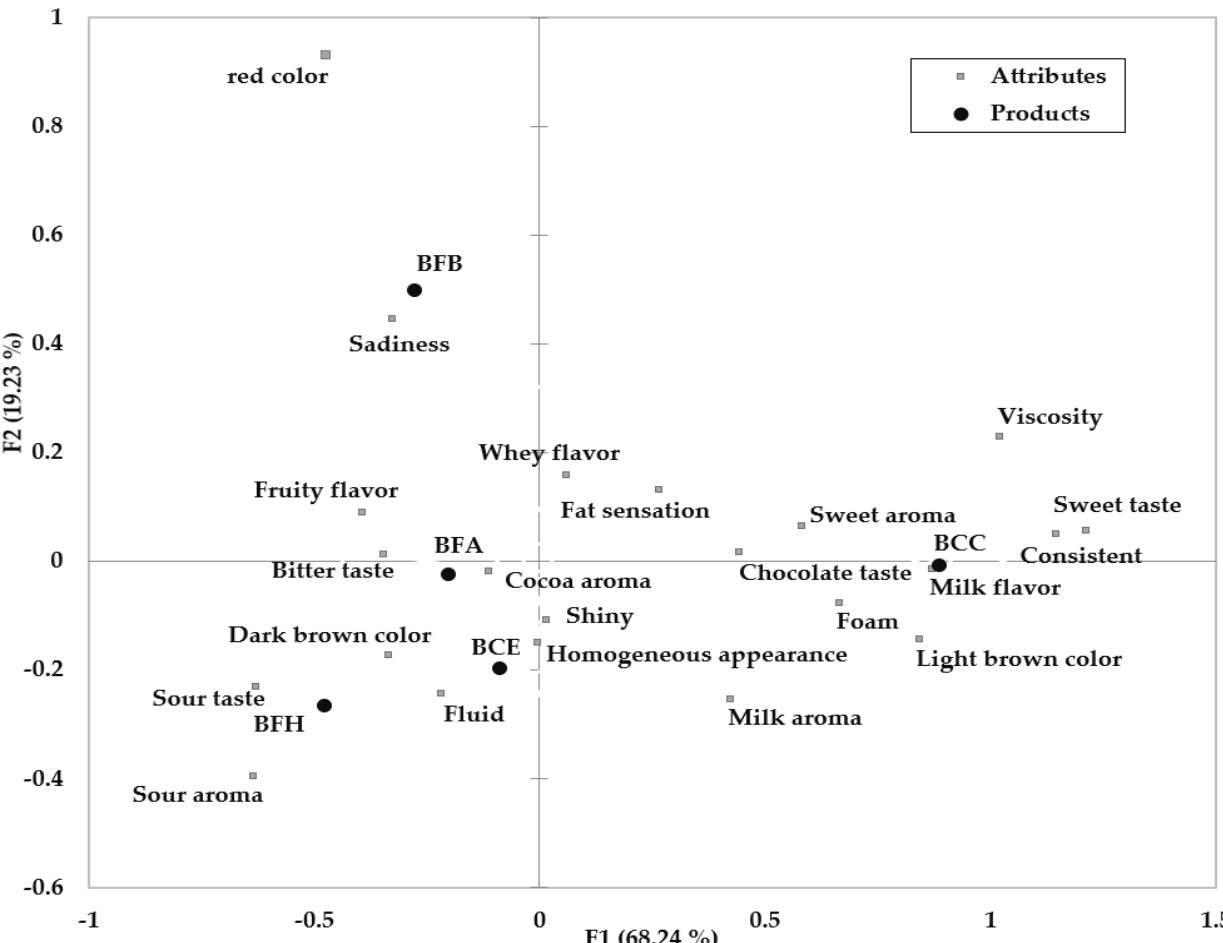

**Figure 2.** Bidimensional representation of principal component analysis (PCA) of buffalo whey-based cocoa beverage. BCC = control commercial formulation; BCE = control experiment formulation; BFA = beverage with açaí flour; BFB = beverage with beetroot flour; BFH = beverage with hibiscus flour.

These findings underscore the critical significance of taking into account sensory attributes and consumer preferences during the development of buffalo whey-based cocoa beverages. A comprehensive understanding of the individual ingredients' influence and their sensory characteristics can be instrumental in fine-tuning formulations, thereby elevating their acceptance and overall popularity among consumers. Furthermore, it is essential to emphasize the importance of incorporating information related to health and sustainability into the product development process. This strategic approach not only aligns products with consumer expectations and desires but also contributes to their market success. Simultaneously, it addresses growing concerns surrounding health and sustainability, ultimately

fostering a more responsible and environmentally conscious industry. In conclusion, the integration of sensory attributes, consumer preferences, and a commitment to health and sustainability is pivotal in the successful creation of cocoa buffalo milk whey-based beverages. This holistic approach not only ensures product acceptance but also positions the industry as a responsible and forward-thinking player in the market.

### 3.4. Consumer Perception

The impact of health and sustainability information on the flavor, sour taste, mouthfeel, overall liking, and purchase intention of the BFA and BFH treatments were not found to be significant ($p < 0.05$). These results suggest that the information provided to consumers did not significantly influence their acceptance of the final product, as indicated in Table 4. This pattern could be attributed to the overall low acceptance of beverages, as previous studies have highlighted the powerful effect of flavor on the acceptance of food products, even when they possess beneficial health characteristics [48].

**Table 4.** The average of the hedonic values obtained from the acceptance test for BFA and BFH treatments by the blind and informed groups.

| Treatment | Information | Attribute | | | | |
|---|---|---|---|---|---|---|
| | | Flavor | Sour Taste | Mouth Feeling | Overall Linking | Purchase Intention |
| BFA [1] | Blind [2] | 6.67 ± 1.88 [a] | 6.78 ± 1.67 [a] | 6.96 ± 1.49 [a] | 6.89 ± 1.67 [a] | 3.50 ± 1.09 [a] |
| | Health [3] | 6.31 ± 1.93 [a] | 6.59 ± 1.76 [a] | 6.56 ± 1.84 [a] | 6.56 ± 1.73 [a] | 3.48 ± 1.08 [a] |
| | Sustainability [4] | 6.17 ± 2.09 [a] | 6.46 ± 2.03 [a] | 6.35 ± 2.04 [a] | 6.22 ± 2.06 [a] | 3.24 ± 1.10 [a] |
| BFH | Blind | 2.98 ± 2.00 [a] | 3.33 ± 1.95 [a] | 3.56 ± 1.85 [a] | 3.37 ± 1.85 [a] | 1.59 ± 0.86 [a] |
| | Health | 3.20 ± 1.94 [a] | 3.19 ± 1.87 | 3.52 ± 2.13 [a] | 3.09 ± 1.85 [a] | 1.61 ± 0.88 [a] |
| | Sustainability | 3.15 ± 2.17 [a] | 3.41 ± 2.33 [a] | 3.20 ± 2.19 [a] | 3.31 ± 2.22 [a] | 1.76 ± 1.04 [a] |

[a] Means in the same column followed by different lowercase letters differ from the kinds of information (Tukey test; $p < 0.05$). [1] BFA = beverage with açaí flour; BFH = beverage with hibiscus flour. [2] Blind = participants received the samples without information. [3] Health = participants received the samples with health information. [4] Sustainability = participants received the samples with sustainability information.

On the other hand, the Q Cochran analysis unveiled a noteworthy distinction ($p < 0.05$) in consumers' emotional responses based on the type of information they received for both treatments. Those who were provided with health-related information were notably more inclined ($p < 0.05$) to employ expressions such as "beneficial for health" and "nutrient-rich beverage" when describing their emotional experience during the trial, as depicted in Table 5. In contrast, sustainability information prompted an increased usage of terms like "sustainable", "supporting nature", and "ecological" for both treatments. Ultimately, these findings underscore the distinct impact of health and sustainability information on consumers' emotional reactions. Health information engendered positive associations with nutritional advantages, while sustainability information sparked an awareness of environmental conscientiousness. Recognizing the influence of such information on consumer emotions can offer invaluable insights for product positioning and communication strategies.

**Table 5.** Frequency (%) of each term of the CATA was indicated by consumers for different information in each buffalo whey-based cocoa beverage.

| Emotion | BFA [1] | | | *p*-Value | BFH | | | *p*-Value |
|---|---|---|---|---|---|---|---|---|
| | Blind [2] | Health [3] | Sustainability [4] | | Blind | Health | Sustainability | |
| Good for health | 37.04 [b] | 70.37 [a] | 31.48 [b] | <0.0001 | 11.11 [b] | 44.44 [a] | 18.52 [b] | 0.000 |
| Nutritious drink | 46.30 [a] | 64.81 [a] | 42.59 [a] | 0.045 | 12.96 [b] | 38.89 [a] | 14.81 [b] | 0.002 |
| Energetic | 29.63 [a] | 18.52 [a] | 24.07 [a] | 0.394 | 9.26 [a] | 11.11 [a] | 11.11 [a] | 0.926 |
| Pacific | 16.67 [a] | 25.93 [a] | 9.26 [a] | 0.079 | 3.70 [a] | 3.70 [a] | 3.70 [a] | 1.000 |
| Happy | 24.07 [a] | 31.48 [a] | 31.48 [a] | 0.633 | 5.56 [a] | 1.85 [a] | 3.70 [a] | 0.607 |
| Optimistic | 33.33 [a] | 37.04 [a] | 27.78 [a] | 0.607 | 5.56 [a] | 7.41 [a] | 7.41 [a] | 0.913 |
| Satisfied | 57.41 [a] | 53.70 [a] | 51.85 [a] | 0.850 | 11.11 [a] | 14.81 [a] | 3.70 [a] | 0.155 |
| Natural drink | 29.63 [a] | 46.30 [a] | 37.04 [a] | 0.166 | 11.11 [a] | 24.07 [a] | 16.67 [a] | 0.186 |
| Sustainable | 16.67 [b] | 20.37 [b] | 55.56 [a] | <0.0001 | 12.96 [b] | 11.11 [b] | 38.89 [a] | 0.000 |
| Ecological | 16.67 [b] | 11.11 [b] | 37.04 [a] | 0.002 | 5.56 [b] | 5.56 [b] | 29.63 [a] | 0.000 |
| Help nature | 9.26 [b] | 16.67 [ab] | 33.33 [a] | 0.004 | 3.70 [b] | 3.70 [b] | 22.22 [a] | 0.001 |
| Animal welfare | 11.11 [a] | 7.41 [a] | 18.52 [a] | 0.193 | 1.85 [a] | 3.70 [a] | 11.11 [a] | 0.050 |
| Animal exploitation | 1.85 [a] | 0.00 [a] | 3.70 [a] | 0.368 | 1.85 [a] | 0.00 [a] | 1.85 [a] | 0.607 |
| Disappointed | 7.41 [a] | 3.70 [a] | 3.70 [a] | 0.607 | 40.74 [a] | 37.04 [a] | 35.19 [a] | 0.823 |
| Rejection | 5.56 [a] | 1.85 [a] | 3.70 [a] | 0.607 | 61.11 [a] | 38.89 [a] | 50.00 [a] | 0.072 |
| Sad | 1.85 [a] | 0.00 [a] | 3.70 [a] | 0.368 | 20.37 [a] | 11.11 [a] | 22.22 [a] | 0.229 |
| Unlike | 7.41 [a] | 1.85 [a] | 5.56 [a] | 0.417 | 55.56 [a] | 48.15 [a] | 42.59 [a] | 0.337 |
| Emotionless | 11.11 [a] | 5.56 [a] | 5.56 [a] | 0.407 | 27.78 [a] | 12.96 [a] | 12.96 [a] | 0.085 |

[a,b] Means in the same line followed by different lowercase letters differ from the kinds of information (Tukey test; $p < 0.05$). [1] BFA = beverage with açaí flour; BFH = beverage with hibiscus flour. [2] Blind = participants received the samples without information. [3] Health = participants received the samples with health information. [4] Sustainability = participants received the samples with sustainability information.

## 4. Conclusions

Higher acceptance levels were observed when cocoa powder was used in lower proportions and combined with cane sugar. This fact highlights the importance of finding the right balance of ingredients to achieve optimal flavor profiles that align with consumer preferences. However, incorporating açaí, beetroot, and hibiscus flour into buffalo whey-based cocoa beverages resulted in a decline in overall acceptance, primarily due to the strong bitter taste associated with hibiscus flour. Furthermore, beverages containing 100% cocoa faced penalties due to inadequate aroma and insufficient sweetness. Remarkably, the type of information provided to consumers significantly impacted their emotional responses. These findings emphasize the role of information in shaping consumer perceptions and highlight the potential of leveraging audiovisual aids, such as videos, to enhance consumer understanding and engagement. Therefore, through optimizing ingredient proportions, addressing flavor concerns, and effectively conveying information, the dairy industry can create products that resonate with consumers, improving acceptance and market success. In this context, further studies can be provided to maximize consumers' acceptance through optimizing formulas with more edulcorates or saccharose and through adding a proportion of whole milk.

**Supplementary Materials:** The following supporting information can be downloaded at: https://www. mdpi.com/article/10.3390/beverages9040090/s1, Table S1: Cow and buffalo whey proximal composition (means ± SD). Table S2: Formulation of cow and buffalo whey-based cocoa beverage. Table S3: Attributes evaluated with definition. Figure S1: Study design illustrating the stages involved in the experiment. BCC = control commercial formulation; BCE = control experimental formulation; BFA = dessert with açaí flour; BFB = dessert with beetroot flour; BFH = dessert with hibiscus flour.

**Author Contributions:** M.P.d.C. and M.J.G.S. conceived the initial idea; M.P.d.C., D.M.O. and M.E.d.O.M. supervised the study; M.J.G.S. and M.P.d.C. developed the methodology and review and wrote the original draft; M.J.G.S., M.P.d.C., A.C.d.O.A., B.S.d.S.R. and U.M.P. carried out the sensory experiments; M.J.G.S., I.L.d.S.R., A.C.d.O.A., B.S.d.S.R., U.M.P., M.P.d.C., D.M.O. and M.E.d.O.M.

collected, analyzed, formatted, and interpreted the data and edited the manuscript. All authors have read and agreed to the published version of the manuscript.

**Funding:** The present study was carried out thanks to the funding of the Conselho Nacional de Desenvolvimento Científico e Tecnológico (grant number. 405728/2018-2, 402430/2018-2, and 303074/2021-3 CNPq, Brazil), the Coordenação de Aperfeiçoamento de Pessoal de Nível Superior (grant number. 88887.507483/2020-00, CAPES, Brazil), the and the Fundação de Amparo à Pesquisa do Estado da Bahia (grant number BOL0444-2020, FAPESB, Brazil).

**Data Availability Statement:** The datasets generated for this study are available on request to the corresponding author.

**Acknowledgments:** The authors would like to thank the Fundação de Amparo à Pesquisa do Estado da Bahia, the Conselho Nacional de Desenvolvimento Científico e Tecnológico, the Coordenação de Aperfeiçoamento de Pessoal de Nível Superior and the Serviço Nacional de Aprendizagem do Cooperativismo (SESCOOP) by the granted.

**Conflicts of Interest:** The authors declare no conflict of interest.

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
