# Peer review of "Buffalo Whey-Based Cocoa Beverages with Unconventional Plant-Based Flours: The Effect of Information and Taste on Consumer Perception"

_beverages, doi:10.3390/beverages9040090_

Round 1

Reviewer 1 Report

Comments and Suggestions for Authors

Introduction:

- The introduction share information about whey in general, but the study uses specifically buffalo whey in the formulations tested. It is not clear anywhere in the manuscript why this choice was made. How representative is this by-product for the dairy industry in Brazil? For example, is there an economical aspect to support this ingredient choice? Please justify this choice.

- Likewise, why did the authors choose to use açaí, beetroot and hibiscus flours?

Lines 41-48: Sentence is too long and confusing. Please rewrite it.

Line 42: What do you mean by “make their vehicles”?

Material and methods:

Lines 68-75: Cow milk and buffalo milk have different composition. Thus, differences in the whey composition are also expected.  Add overall composition of the buffalo whey (protein, water, etc.) used

Lines 69-70: add reference for the coalho cheese production methodology.

Line 70: detail the filtration process.

Line 80-81: What are these “typical market features”? please clarify

Lines 76-91: What was the percentage of buffalo whey used in the products? Please add a table with all the formulations. It will make it easier to understand and compare differences among controls and tested products.

Line 99: Replace “Firstly” by “First”

Line 101: Replace “Secondly” by “Second” or “Then”

Lines 102-104: The population profile for each study is not clear. Please separate study 1 from study 2 as the participants were not the same for both studies.

Lines 150-151: “based on whey” doesn’t make much sense. Would it be “made with whey” a better translation?

Line 156: Does “taste” here represent “overall taste”?  if yes, please clarify.

Results:

Figure 1: Numbers 1 and 2 are not present in the image. Please include.

Table 2: include the percentage symbol (%) and place percentage of consumers and mean drop in two separate columns

Figure 2: change color red to blue, for example, to make the plot more color-blind friendly.

There was no discussion about sensory characteristics linked to buffalo whey, which is a less common ingredient than cow milk whey.

Conclusions:

Your first conclusion is about cocoa powder and it was not part of the objectives. Why did the authors use different cocoa powders? Please review this section and the description of the study objectives. Your conclusions should be coherent with your research questions.

Comments on the Quality of English Language

Overall comment (applies for the whole manuscript): Break down the sentences in shorter statements to make the reading clearer. Although the very long sentences are common in Portuguese, most of the time they don’t read well in English and tend to be confusing. Also, give preference to direct sentences.

Author Response

GENERAL COMMENTS BY THE AUTHORS:

We would like to thank the reviewers for their thoughtful critique of our manuscript. We believe that we have fully addressed all of these concerns and comments, which has increased the overall impact of the manuscript.

All modifications were highlighted in yellow.

RESPONSE TO REVIEWER:

Author's Reply to the Review Report

Comments and Suggestions for Authors:

The manuscript contains interesting findings but requires additional corrections, which will enable an even more comprehensive approach to the described topic. Below are my proposals, in order to refine the manuscript.

Answer: We sincerely appreciate each of your insightful comments, as they have greatly contributed to enhancing the quality of our study. To address these valuable suggestions, we have made several important revisions to our work.

Firstly, we have restructured our introduction by incorporating additional pertinent information. This serves to offer greater clarity in our justification for the study and a more comprehensive explanation of our objectives. Furthermore, we have expanded upon the details of our methodology, particularly with regard to the production process. These modifications aim to ensure that any reader can readily comprehend the products upon which our study was conducted. Additionally, we have refined the wording of our methodology to make each step more accessible for understanding. We have also introduced tables and figures to facilitate comprehension at various stages of our study.

In parallel, we have conducted a more thorough exploration of our results in an effort to provide a more robust explanation of the impact of different flour types on consumers' perception. This approach enables us not only to discern the individual effects of each flour but also to pave the way for future projects, allowing for optimization in both formulation and processing. Thank you for your valuable feedback, which contributed to the strength of our publication.

Introduction:

- The introduction share information about whey in general, but the study uses specifically buffalo whey in the formulations tested. It is not clear anywhere in the manuscript why this choice was made. How representative is this by-product for the dairy industry in Brazil? For example, is there an economical aspect to support this ingredient choice? Please justify this choice.

Answer: Thank you for your concern. While we acknowledge that buffalo whey may not represent the entire dairy industry in Brazil, our research was designed to address a specific set of questions related to this particular by-product, considering its unique composition and properties. We believed that by focusing on buffalo whey, we could provide specific insights that may not apply to more commonly used whey types. To clarify it we added an explanation to the introduction.

- Likewise, why did the authors choose to use açaí, beetroot and hibiscus flours?

Answer: Thank you for your inquiry. The selection of açaí, beetroot, and hibiscus flours was driven by a combination of factors, including their nutritional profile, consumer interest in health-promoting foods, their unique sensory characteristics, and considerations of sustainability. These ingredients were chosen to create a dessert that not only tasted great but also contributed positively to health and sustainability goals. We improved this section to provide clarity.

Lines 41-48: Sentence is too long and confusing. Please rewrite it.

Answer: Thank you for your comment, the sentence was reformulated and were tried to make the reading easier and understanding.

Line 42: What do you mean by “make their vehicles”?

Answer: Thank you, we suggest that desserts made with whey can be used as vehicles for important bioactive components. Also, the sentences were reformulated for doing easier to reading understood or point of view.

Material and methods:

Lines 68-75: Cow milk and buffalo milk have different composition. Thus, differences in the whey composition are also expected. Add overall composition of the buffalo whey (protein, water, etc.) used.

Answer: Thank you for your consideration. The proximal composition analyzes in cow and buffalo whey were carried out and were placed in a table (Table S1), as suggested, to have a better visualization of the differences. The Automatic milk analyzer (Model LACTOSCAN SP) was used to perform the analyzes.

Lines 69-70: add reference for the coalho cheese production methodology.

Answer: Thank you for your comment. We used the methodology described by Rekowsky et al. (2022), we believe your observation is important, in this sense the added information.

Rekowsky, B. S. D. S.; Monteiro, M. L. G.; Silva, T. M.; Conté-Júnior, C. A.; Costa, M. P. D. Semi-hard buffalo cheese: how cow's milk affects sensory acceptance?. Brazilian Journal of Food Technology 2022, 25. https://doi.org/10.1590/1981-6723.03022

Line 70: detail the filtration process.

Answer: Thank you for your suggestion, as mentioned in the text, the whey filtration was carried out using fabric material, to prevent found cheese particles in the whey that could cause alterations in the chocolate beverage.

Line 80-81: What are these “typical market features”? please clarify

Answer: Thank you, we mean the typical formulations found on the market, that is, those chocolate desserts from the most consumed brands in Brazil. The information has been reformulated to improve the understanding of the reader.

Lines 76-91: What was the percentage of buffalo whey used in the products? Please add a table with all the formulations. It will make it easier to understand and compare differences among controls and tested products.

Answer: Thank you. The information has been included in the same paragraph. Additionally, the information has been presented in table (Table S2), as recommended, as supplementary material to enhance the reader's comprehension.

Line 99: Replace “Firstly” by “First”

Answer: Thank you, the information was revised and modified as suggested.

Line 101: Replace “Secondly” by “Second” or “Then”

Answer: Thank you, the information was revised and modified as suggested.

Lines 102-104: The population profile for each study is not clear. Please separate study 1 from study 2 as the participants were not the same for both studies.

Answer: Thank you for your comment, the information was revised and separated as suggest. For better understood in the first section, the acceptability and characterization of the cow and buffalo whey-based chocolate beverage were evaluated through overall acceptability, purchase intention, and JAR (Just-About-Right) scaling, was conducted at the School of Veterinary Medicine of the Federal University of Bahia, Brazil, involved a sample size of 120 untrained participants (84 women and 36 men, respectively), with ages ranging from 18 to 66 years old.

In other ways, the second study an assessment of the effect of information was conducted. The analyses were conducted at the Faculty of Pharmacy, Nutrition School, and School of Veterinary Medicine of the Federal University of Bahia, Brazil. The study involved a sample size of 164 untrained participants (114 women and 50 men, respectively), with ages ranging from 18 to 66 years old.

Lines 150-151: “based on whey” doesn’t make much sense. Would it be “made with whey” a better translation?

Answer: Thank you, the information was revised and modified as suggested.

Line 156: Does “taste” here represent “overall taste”?  if yes, please clarify.

Answer: Thank you for your comment. The information has been reviewed and modified as suggested. Additionally, a table that was used to assist our assessors in understanding the description of the attributes has been added as supplementary material (Table S3).

Results:

Figure 1: Numbers 1 and 2 are not present in the image. Please include.

Answer: Thank you for your comment, the figures have been corrected to improve their visualization.

Table 2: include the percentage symbol (%) and place percentage of consumers and mean drop in two separate columns

Answer: Thank you. We attempted to follow the suggested approach by placing the percentage mean drop in separate columns, but we realized that it wasn't as clear and easy to interpret. Therefore, we have made modifications to Tables 1 and 2 by arranging the treatments on the top (horizontal) and the attributes below (vertically). This adjustment should facilitate the reading and interpretation of the presented data.

Figure 2: change color red to blue, for example, to make the plot more color-blind friendly.

Answer: Thank you for your observation, the information was revised and the plot visual has changed to be easier to interpret.

There was no discussion about sensory characteristics linked to buffalo whey, which is a less common ingredient than cow milk whey.

Answer: Thank you for your concern. We agree with the lack of discussion about our based-whey, however, as has been shown throughout the study, our objective was not to mark differences between cow and buffalo whey-based chocolate beverage. This would require more specific sensory analysis and with trained participants with enough knowledge in distinguish small changes in flavor, due the beverages were made with whey, a by-product of milk, it is more difficult to distinguish the differences between cow and buffalo whey.

Conclusions:

Your first conclusion is about cocoa powder and it was not part of the objectives. Why did the authors use different cocoa powders? Please review this section and the description of the study objectives. Your conclusions should be coherent with your research questions.

Answer: Thank you for your observation, in the first section of the study, it was proposed to bring a chocolate beverage as a reference that reflects the product it can be found in the dairy beverages market, in this sense it was chosen to use 50 % cocoa powder, which is the most used in the formulations of chocolate beverage. In the case of 100% cocoa, it was chosen for its nutritional value, with a greater content of phenolic compounds. The justification has been added to the introduction to provide more clarity for the study.

Comments on the Quality of English Language

Overall comment (applies for the whole manuscript): Break down the sentences in shorter statements to make the reading clearer. Although the very long sentences are common in Portuguese, most of the time they don’t read well in English and tend to be confusing. Also, give preference to direct sentences.

Answer: Thank you for your feedback regarding the quality of English language in our manuscript. We believe we have significantly enhanced the clarity of our manuscript and reduced potential confusion for English-speaking readers. We also gave preference to direct sentences throughout the manuscript, ensuring that our writing was straightforward and to the point.

Reviewer 2 Report

Comments and Suggestions for Authors

1.    Quite few literature items, perhaps they should expand the discussion and refer to other publications regarding the methodology

2.    Line 33: I suggest adding the word by-products to your keywords

3.    Line 63: Instead of [10, 11, 12, 13] do [10-13]

4.    Line 71: how long was it refrigerated?

5.    Line 90-91: Refrigerated again, for how long? Were they safe (no information about the durability of the product)

6.    Line 108: What was the storage temperature? One paragraph has in 8°C and the other has 4°C

7.    Line 135: Instead of “perception s” should be “perceptions”

8.    Line 166: Shouldn't a non-parametric analysis be carried out in the case of this type of (spike?) values?

9.    Line 171: “(P < 0.05)” should be p<0.05, the same in line 184, 187, 190, 191, 194 but line 296, 309, 311  in lower letter– please standardize it throughout the work

10.  line 209: Figure 1 – caption on the chart: “overall linking” intead of “overall liking”

abbreviations indicating sample legend have B instead of the first letter D, e.g. BCC, BCE.

There are superscripts 1 and 2 in the figure caption - but there is no reference to them in the chart … (1DCC = control commercial formulation; DCE = con- 213 trol experimental formulation; DFA = dessert with açai flour; DFB = dessert with beetroot flour; DFH = dessert with hibiscus flour. 2 Evaluated on a 5-point category scale)

no X and Y axes caption on the chart

If a 9-point 115 hedonic scale was used (from 1 = extremely dislike to 9 = extremely like), why do the values in the chart end with the number 7?

11.  Line 222: After formulating […] as noted by de Jesus Silva et al. [19] place a dot (.) instead of a comma (,).

12.  Line 281: Please correct the resolution of figure 2

13.  Line 282: Figure 2 – figure caption – adjust the font size to the requirements of the magazine, maintain a space between the figure caption and the text of the work

14.  Line 364-428: References should be described as presented in the Instructions for Authors, depending on the type of work

Author Response

GENERAL COMMENTS BY THE AUTHORS:

We would like to thank the reviewers for their thoughtful critique of our manuscript. We believe that we have fully addressed all of these concerns and comments, which has increased the overall impact of the manuscript.

All modifications were highlighted in yellow.

RESPONSE TO REVIEWER:

Comments and Suggestions for Authors

  1. Quite few literature items, perhaps they should expand the discussion and refer to other publications regarding the methodology.

Answer: We sincerely appreciate each of your insightful comments, as they have greatly contributed to enhancing the quality of our study. To address these valuable suggestions, we have made several important revisions to our work.

Firstly, we have restructured our introduction by incorporating additional pertinent information. This serves to offer greater clarity in our justification for the study and a more comprehensive explanation of our objectives. Furthermore, we have expanded upon the details of our methodology, particularly with regard to the production process. These modifications aim to ensure that any reader can readily comprehend the products upon which our study was conducted. Additionally, we have refined the wording of our methodology to make each step more accessible for understanding. We have also introduced tables and figures to facilitate comprehension at various stages of our study.

In parallel, we have conducted a more thorough exploration of our results in an effort to provide a more robust explanation of the impact of different flour types on consumers' perception. This approach enables us not only to discern the individual effects of each flour but also to pave the way for future projects, allowing for optimization in both formulation and processing. Thank you for your valuable feedback, which contributed to the strength of our publication.

  1. Line 33: I suggest adding the word by-products to your keywords

Answer: Thank you for your comment, we included the keyword as suggested.

  1. Line 63: Instead of [10, 11, 12, 13] do [10-13]

Answer: Thank you for your comment, the information was revised and modified as suggested. We use the comma only in cases that are not consecutive numbers.

  1. Line 71: how long was it refrigerated?

Answer: Thank you, the products were refrigerated for just one day at 4 ± 2 °C. Usually, the whey-based chocolate beverage and next after the sensory analyses were carry out. The information was added in the text for improved the reader understood.

  1. Line 90-91: Refrigerated again, for how long? Were they safe (no information about the durability of the product)

Answer: Thank you, the products were refrigerated for just one day and microbiological analysis was carried out on the desserts to guarantee the quality of the products, following Brazilian regulations. The information has been added to the text to provide more clarity in the methodology.

  1. Line 108: What was the storage temperature? One paragraph has in 8°C and the other has 4°C

Answer: Thank you for your comment, we usually refrigerate our products in a refrigerator that can vary its temperature, but we try to maintain a temperature of 4 ± 2 °C. Therefore, the information was corrected in the text to provide more clarity in the methodology.

  1. Line 135: Instead of “perception s” should be “perceptions”

Answer: Thank you, the information was revised and modified as suggested.

  1. Line 166: Shouldn't a non-parametric analysis be carried out in the case of this type of (spike?) values?

Answer: Thank you for your concern. ANOVA is a common statistical method used in sensory analysis to analyze data obtained from sensory evaluations. In fact, ANOVA can be used to analyze this data by comparing the means of the sensory attributes across different products or treatments to determine if there are significant differences between them. In this study, we used one-way ANOVA to compare the means of the sensory attributes across different treatments. It’s worth noting that sensory analysis data can sometimes have non-normal distributions or unequal variances, as the reviewer stated, which can violate the assumptions of traditional ANOVA. In such cases, alternative statistical methods such as non-parametric tests or data transformation may be necessary. To confirm, we cited studies that have been published in Beverages applying ANOVA to sensory data.

Sandilya, S.; Oroya, N.; Moral, T.; Vázquez-Araújo, L. Effect of Recipient’s Tactile Properties and Expectations on Beer Perception. Beverages 2023, 9, 75. https://doi.org/10.3390/beverages9030075

Franceschi, D.; Lomolino, G.; Sato, R.; Vincenzi, S.; De Iseppi, A. Umami in Wine: Impact of Glutamate Concentration and Contact with Lees on the Sensory Profile of Italian White Wines. Beverages 2023, 9, 52. https://doi.org/10.3390/beverages9020052

Bendaali, Y.; Vaquero, C.; Escott, C.; González, C.; Morata, A. Isotonic Drinks Based on Organic Grape Juice and Naturally Flavored with Herb and Spice Extracts. Beverages 2023, 9, 49. https://doi.org/10.3390/beverages9020049

  1. Line 171: “(P < 0.05)” should be p<0.05, the same in line 184, 187, 190, 191, 194 but line 296, 309, 311 in lower letter– please standardize it throughout the work

Answer: Thank you for the observation, the information was revised throughout the manuscript.

  1. line 209: Figure 1 – caption on the chart: “overall linking” instead of “overall liking”

Answer: Thank you for the observation, we revised the information and corrected the spelling.

Abbreviations indicating sample legend have B instead of the first letter D, e.g. BCC, BCE.

Answer: Thank you, the information was revised and the letters have been corrected to be correct and standardized as established.

There are superscripts 1 and 2 in the figure caption - but there is no reference to them in the chart … (1DCC = control commercial formulation; DCE = con- 213 trol experimental formulation; DFA = dessert with açai flour; DFB = dessert with beetroot flour; DFH = dessert with hibiscus flour. 2 Evaluated on a 5-point category scale)

Answer: Thank you for your comment, the information was revised and the superscripts were removed, counting that are not need for any context.

No X and Y axes caption on the chart

Answer: Thank you for your comment, the information was revised and added the X and Y axes information, in this way, Attributes are located in the axes X and the score for each product for each attribute in the axes Y.

If a 9-point 115 hedonic scale was used (from 1 = extremely dislike to 9 = extremely like), why do the values in the chart end with the number 7?

Answer: Thank you for the observation. The information has been revised. Since the highest score we found was below 7, we initially considered keeping it within the range of 0 to 7. However, after reviewing the chart, we concluded that it would be easier to understand if we expanded the range from 1 to 9. This decision was based on the assessment measures used in the methodology.

  1. Line 222: After formulating […] as noted by de Jesus Silva et al. [19] place a dot (.) instead of a comma (,).

Answer: Thank you, the information was revised and modified as suggested

  1. Line 281: Please correct the resolution of figure 2

Answer: Thank you for your consideration. The resolution of the figures was corrected and improve, as suggested, to provide more clarity.

  1. Line 282: Figure 2 – figure caption – adjust the font size to the requirements of the magazine, maintain a space between the figure caption and the text of the work

Answer: Thank you, the information was revised and corrected according to journal standards.

  1. Line 364-428: References should be described as presented in the Instructions for Authors, depending on the type of work

Answer: Thank you, the information was revised and corrected according to journal standards.

Reviewer 3 Report

Comments and Suggestions for Authors

Review Beverages 2630264

The manuscript contains interesting findings but requires additional corrections, which will enable an even more comprehensive approach to the described topic. Below are my proposals, in order to refine the manuscript:

Line 2 title

I suggest to add in the title information about type of flours used.

Line 25

Why only that desserts were taken to the second stage?

Line 67

I suggest to ad a graph with explanation step by step what was done in the experiment and what kind of analysis were made for which samples. It is not clear now.

Line 68-69

Please add references for methodology

Line 69

Please explant what it is coalho cheese to the readers

Line 72 is the product that you created soli od liquid

Please provide pictures for better understanding the idea of the product.

Line 74

Please add specification for the used ingredients, name of producers, what type of inulin, etc

Line 77

Did you use powder or liquid whey?

Line 84-91

Please provide the table with the formula tat was used in experiment (ingredient percentage etc) It is not clear now

Line 88

Please write how much whey did you add in each formula. Statement in the manuscript is not sufficient.

Line 91

What about pasterisation of commercial product and prepared one?

Line 103

Where these participant trained or untrained?

Line 103

How many participants have you used for each method?

Line 112

How many samples you were evaluated in each method?

Line 155

Add references for the method

Line 126

How did you train people for such a method?

Line 127

Please add the list of attributes with definition in the manuscript or the attachment

Line 139

Why only that desserts were taken to the second stage?

Line 135

Please add the list of attributes with definition in the manuscript or the attachment

Line 176

There is lack of discussion in the manuscript. Please add it to each results that are explained.

Line 226 Tabel 1 and 2

Please modify the table. Put treatment on the top and attributes below, it will be clear to compere the results.

Line 328

Deep explanation of influence and role of each flour in used methods is needed.

Author Response

GENERAL COMMENTS BY THE AUTHORS:

We would like to thank the reviewers for their thoughtful critique of our manuscript. We believe that we have fully addressed all of these concerns and comments, which has increased the overall impact of the manuscript.

All modifications were highlighted in yellow.

RESPONSE TO REVIEWER:

Comments and Suggestions for Authors

The manuscript contains interesting findings but requires additional corrections, which will enable an even more comprehensive approach to the described topic. Below are my proposals, in order to refine the manuscript.

Answer: We sincerely appreciate each of your insightful comments, as they have greatly contributed to enhancing the quality of our study. To address these valuable suggestions, we have made several important revisions to our work.

Firstly, we have restructured our introduction by incorporating additional pertinent information. This serves to offer greater clarity in our justification for the study and a more comprehensive explanation of our objectives. Furthermore, we have expanded upon the details of our methodology, particularly with regard to the production process. These modifications aim to ensure that any reader can readily comprehend the products upon which our study was conducted. Additionally, we have refined the wording of our methodology to make each step more accessible for understanding. We have also introduced tables and figures to facilitate comprehension at various stages of our study.

In parallel, we have conducted a more thorough exploration of our results in an effort to provide a more robust explanation of the impact of different flour types on consumers' perception. This approach enables us not only to discern the individual effects of each flour but also to pave the way for future projects, allowing for optimization in both formulation and processing. Thank you for your valuable feedback, which contributed to the strength of our publication.

Line 2 title - I suggest to add in the title information about type of flours used.

Answer: Thank you, the information was revised and modified as suggested, we have been modified the title for “Buffalo whey-based chocolate beverages with unconventional plant-based flours: effect of information and taste in consumer perception”

Line 25 - Why only that desserts were taken to the second stage?

Answer: Thank you, as shown in figure 1 with the results of the acceptance test, the treatments with açaí flour and hibiscus flour, BFA and BFH, respectively, had greater and lesser acceptance, respectively. They were selected mainly to observe whether the type of product could increase the effect of information on the perception of products.

Line 67 - I suggest to ad a graph with explanation step by step what was done in the experiment and what kind of analysis were made for which samples. It is not clear now.

Answer: Thank you for your observation. We believe it is of great importance to make our study easier to read and to help readers understand the methodology used in the study. Therefore, we have created a graph that simplifies the explanation of the methodology used in our study. To achieve this, we have included Figure S1 as supplementary material.

Line 68-69 - Please add references for methodology

Answer: Thank you for your comment. We used the methodology described by Rekowsky et al. (2022), we believe your observation is important, in this sense the added information.

Rekowsky, B. S. D. S.; Monteiro, M. L. G.; Silva, T. M.; Conté-Júnior, C. A.; Costa, M. P. D. Semi-hard buffalo cheese: how cow's milk affects sensory acceptance?. Brazilian Journal of Food Technology 2022, 25. https://doi.org/10.1590/1981-6723.03022

Line 69 - Please explant what it is coalho cheese to the readers

Answer: Thank you, we included the information and was add the respective references that has been used for the coalho cheese production.

Line 72 - is the product that you created soliod liquid. Please provide pictures for better understanding the idea of the product.

Answer: Thank you for your comment, the products created in the study were in liquid form. Given their physical-chemical characteristics and upon further examination of your question, we determined it would be more appropriate to refer to them as chocolate drinks made from buffalo whey. Additionally, we have included Figure S1 as supplementary material to enhance clarity.

Line 74 - Please add specification for the used ingredients, name of producers, what type of inulin, etc

Answer: Thank you for your comment, we included the principal information about the ingredients: Açai flour (Viva Natureza®, Bahia, Brazil), xanthan gum (Leve Crock®, Piraí do Sul, Brazil), xylitol (Natural Vitta®, Bahia, Brazil), and cocoa powder (50% and 100%) (Nestlé Dos Frades®, Brazil), were purchased from a local natural product store in Salvador, Bahia, Brazil. Additionally, inulin (Sweetmix®, Sorocaba, Brazil), beetroot powder (Natural Vitta®, Bahia, Brazil), and hibiscus flour (Della Terra, São Paulo, Brazil).

Line 77 - Did you use powder or liquid whey?

Answer: Thank you for your concern, the whey used in the preparation of the drinks was in liquid form, a byproduct of coalho cheese production. The whey was separated and pasteurized for use in the production of the beverages.

Line 84-91 - Please provide the table with the formula tat was used in experiment (ingredient percentage etc) It is not clear now

Answer: Thank you. The information has been included in the same paragraph. Additionally, the information has been presented in table (Table S2), as recommended, as supplementary material to enhance the reader's comprehension.

Line 88 - Please write how much whey did you add in each formula. Statement in the manuscript is not sufficient.

Answer: Thank you for your comment, the amount of serum depended on the type of treatment, we worked with 3 types of formulations, consequently, for the BCC treatment 88.75 % was used, for the BCE it was 86.5 % and for the BFA, BFB and BFH treatments 84.0 % was used. However, the information has been presented in table (Table S2), as recommended, as supplementary material to enhance the reader's comprehension.

Line 91 - What about pasterisation of commercial product and prepared one?

Answer: Thank you, the products available on the market are sold in sealed and vacuum packaging. Furthermore, microbiological analysis was conducted on the desserts to ensure the quality of the products, in compliance with Brazilian regulations. This information has been added to emphasize the hygiene aspect of the products.

Line 103 - Where these participant trained or untrained?

Answer: Thank you for your comment, we included this information to provide clarity. In this study, preference analyzes were carried out which do not require trained people to be carried out. However, verbal information was provided on how to perform sensory analysis, in addition to presenting texture information that facilitated the understanding of each participant.

Line 103 - How many participants have you used for each method?

Answer: Thank you for you observation, we included this information and separated by section. The first section involved a sample size of 120 untrained participants (84 women and 36 men, respectively), with ages ranging from 18 to 66 years old, and the second involved a sample size of 164 untrained participants (114 women and 50 men, respectively), with ages ranging from 18 to 66 years old.

Line 112 - How many samples you were evaluated in each method?

Answer: Thank you for you observation, we included this information and separated by section. In the first section, we used five samples for study the acceptability and characterization of the cow and buffalo whey-based chocolate beverage were evaluated through overall acceptability, purchase intention, and JAR (Just-About-Right) scaling. For the second section were evaluated the result from the figure 1, acceptance test, and were selected the treatment with açaí flour and hibiscus flour, BFA and BFH, respectively, due they showed the greater and lesser acceptance, respectively. After that, the effect of information was conducted using the two samples, BFA and BFH, and other samples, containing a mixture of both.

Line 155 - Add references for the method

Answer: Thank you, we include this information. The methodology was performance according to Costa et al. (2017) with some modification. In this sense, perception analysis was carried out using 3 types of information, health, sustainability and uninformed.

Costa, M. P.; Monteiro, M. L. G.; Frasao, B. S.; Silva, V. L. M.; Rodrigues, B. L., Chiappini, C. C. J.; Conte-Junior, C. A. Consumer perception, health information, and instrumental parameters of cupuassu (Theobroma grandiflorum) goat milk yogurts. Journal of Dairy Science 2017, 100(1), 157–168. https://doi.org/10.3168/jds.2016-11315

 Line 126 - How did you train people for such a method?

Answer: Thank you, in this study all the participants were untrained. we include this information.

Line 127 - Please add the list of attributes with definition in the manuscript or the attachment

Answer: Thank you, the information has been reviewed. A table that was used to assist our assessors in understanding the description of the attributes has been added as supplementary material (Table S3).

Line 139 - Why only that desserts were taken to the second stage?

Answer: Thank you, as shown in figure 1 with the results of the acceptance test, the treatments with açaí flour and hibiscus flour, BFA and BFH, had greater and lesser acceptance, respectively. They were selected mainly to observe whether the type of product could increase the effect of information on the perception of products.

Line 135 - Please add the list of attributes with definition in the manuscript or the attachment

Answer: Thank you, the studies were carried out with random participants who had no training, the definition of the attributes were in table that was used to assist our assessors in understanding the description of the attributes has been added as supplementary material.

Line 176 - There is lack of discussion in the manuscript. Please add it to each results that are explained.

Answer: Thank you very much for the comment, we agree with the observation regarding the lack of discussion about our presented results. In this way, we conducted a more comprehensive analysis of the data presented in our tables and figures, with the aim of identifying significant differences that hold substantial relevance within the context of our findings. Moreover, we performed correlations between the outcomes of our analyses and the composition of the ingredients employed in the formulation of the beverages. This approach allowed us to gain a deeper insight into the influence of different flour types on the overall acceptance of the beverages and how these products were perceived by the study participants.

Line 226 Tabel 1 and 2 - Please modify the table. Put treatment on the top and attributes below, it will be clear to compere the results.

Answer: Thank you for your comment, we have made modifications to Tables 1 and 2 by arranging the treatments on the top (horizontal) and the attributes below (vertically). This adjustment should facilitate the reading and interpretation of the presented data.

Line 328 - Deep explanation of influence and role of each flour in used methods is needed.

Answer: Thank you very much for the comment, we conducted a comprehensive analysis of our findings, aimed at achieving a deeper understanding of the influence of beverages on consumer acceptance and perception. In this regard, acceptance analyses played a pivotal role in determining the levels of acceptance associated with the beverages and assessing the impact of different flour types on that acceptance. Additionally, the Just-About-Right (JAR) analyses facilitated an insight into how consumers perceived the concentrations of ingredients, thereby providing valuable insights for process optimization and enhancing beverage acceptance. Furthermore, in combination with the Check-All-That-Apply (CATA) analyses, we examined the correlation between attributes and the beverages, a crucial aspect for optimization purposes. Most notably, we scrutinized the primary characteristics of each type of flour as perceived by consumers.

Round 2

Reviewer 1 Report

Comments and Suggestions for Authors

Thank you for addressing all previous comments. The introduction section is much more complete and the manuscript reads much better.

Author Response

GENERAL COMMENTS BY THE AUTHORS:

RESPONSE TO REVIEWER:

Author's Reply to the Review Report

Thank you for addressing all previous comments. The introduction section is much more complete and the manuscript reads much better.

Answer: We would like to express our sincere gratitude for your valuable feedback. Each of your suggestions has played a pivotal role in enhancing the overall writing quality of our study.

Reviewer 3 Report

Comments and Suggestions for Authors

Thank you for your explanantion and all your chnages in the manucript that made it better for future readers.

After reading your additional data, I suggest to change the title and all the data in the text into “Buffalo whey-based cocoa beverages with unconventional plant-based flours: effect of information and taste in consumer perception”

There is cocoa beverage not chocolate, becuse you did not add chocolate ingredents to your drink but you have added only cocoa. 

Author Response

GENERAL COMMENTS BY THE AUTHORS:

RESPONSE TO REVIEWER:

Author's Reply to the Review Report

Thank you for your explanantion and all your chnages in the manucript that made it better for future readers.

After reading your additional data, I suggest to change the title and all the data in the text into “Buffalo whey-based cocoa beverages with unconventional plant-based flours: effect of information and taste in consumer perception”

There is cocoa beverage not chocolate, becuse you did not add chocolate ingredents to your drink but you have added only cocoa.

Answer: We would like to express our sincere gratitude for your valuable feedback. Each of your suggestions has played a pivotal role in enhancing the overall writing quality of our study. We be in accord with your suggestion to replace the term “chocolate” with “cocoa” as it constitutes the primary ingredient in the formulation employed
